# Wnt Signaling in Ovarian Cancer Stemness, EMT, and Therapy Resistance

**DOI:** 10.3390/jcm8101658

**Published:** 2019-10-11

**Authors:** Miriam Teeuwssen, Riccardo Fodde

**Affiliations:** Department of Pathology, Erasmus MC Cancer Institute, Erasmus University Medical Center, 3015 GD Rotterdam, The Netherlands

**Keywords:** Ovarian cancer, Wnt signaling, cancer stem cells, tumor progression, therapy resistance, exosomes

## Abstract

Ovarian cancers represent the deadliest among gynecologic malignancies and are characterized by a hierarchical structure with cancer stem cells (CSCs) endowed with self-renewal and the capacity to differentiate. The Wnt/β-catenin signaling pathway, known to regulate stemness in a broad spectrum of stem cell niches including the ovary, is thought to play an important role in ovarian cancer. Importantly, Wnt activity was shown to correlate with grade, epithelial to mesenchymal transition, chemotherapy resistance, and poor prognosis in ovarian cancer. This review will discuss the current knowledge of the role of Wnt signaling in ovarian cancer stemness, epithelial to mesenchymal transition (EMT), and therapy resistance. In addition, the alleged role of exosomes in the paracrine activation of Wnt signaling and pre-metastatic niche formation will be reviewed. Finally, novel potential treatment options based on Wnt inhibition will be highlighted.

## 1. Introduction

Epithelial ovarian cancer (EOC) represents the deadliest among gynecologic malignancies [1]. This is mainly due to the fact that up to 80% of ovarian cancer patients present with symptoms and are subsequently diagnosed only at late disease stages, i.e., when metastases have already spread to pelvic organs (stage II), the abdomen (stage III), or beyond the peritoneal cavity (stage IV) [2]. 

EOC is an extremely heterogeneous disease. Multiple (epi)genetic alterations at a broad spectrum of oncogenes and tumor suppressor genes have been observed in ovarian cancer leading to deregulation of signal transduction pathways whose functions ranges from DNA repair, cell proliferation, apoptosis, cell adhesion, and motility. Based on these molecular alterations, ovarian cancer has been subdivided in two major type I and type II classes of tumors [3]. Type I tumors are slow growing, mostly restricted to the ovary, and develop from well-established precursor lesions called “borderline” tumors. Type I tumors comprise of four different subtypes, namely low-grade serous, mucinous, clear cell, and endometrioid cancers. The histological composition of these four types resemble normal cells present in the fallopian tube and/or ovarian surface epithelium, endocervix, vagina, and endometrium, respectively, thus suggesting different cells of origin for the different histotypes [3]. Type I lesions frequently carry mutations in *KRAS*, *BRAF*, *PTEN*, and *CTNNB1* (β-catenin), and often show a relatively stable karyotype. 

Type II ovarian cancers include high-grade serous (HGS) and undifferentiated carcinomas, the vast majority of which characterized by *TP53* alterations and pronounced genomic instability [3]. Of note, inherited and somatic *BRCA1* and *BRCA2* mutations are usually found in type II tumors. It is under debate whether HGS ovarian cancers originate from the fimbria of the fallopian tube or from the ovarian surface epithelium (OSE) [4].

Ovarian cancers are thought, because of their distinctive progression and recurrence patterns, to be characterized by a hierarchical structure with cancer stem cells (CSCs) endowed with self-renewal and the capacity to differentiate, which continuously fuel the growth of the tumor mass and coexist with more committed cell types [5,6]. Notably, the Wnt/β-catenin signaling pathway, known to regulate stemness in a broad spectrum of stem cell niches including the ovary, is thought to play an important role in ovarian cancer. First, 16–54% of endometrioid ovarian cancers are characterized by mutations in β-catenin or, though at a considerably less frequency, in other members of the Wnt cascade such as *APC*, *AXIN1*, and *AXIN2* [7,8]. Second, other histotypes, and in particular serous ovarian carcinomas where mutations in Wnt-related genes are relatively uncommon, are characterized by constitutive Wnt signaling activation as indicated by alterations in β-catenin subcellular localization (i.e., nuclear and cytoplastic vs. membrane-bound) [9,10,11,12]. Importantly, Wnt activity was shown to correlate with grade [12], epithelial to mesenchymal transition (EMT) [7], chemo-resistance [13], and poor prognosis [14] in patients with ovarian carcinomas. 

Here, we will review the current knowledge of the role of Wnt signaling in ovarian cancer stemness, EMT, and therapy resistance. The alleged role of exosomes in the paracrine activation of Wnt signaling, and novel potential treatment options based on Wnt inhibition will also be highlighted.

## 2. The Wnt/β-Catenin Signaling Pathway

Stem cells are distinguished from other somatic cells by their ability to self-renew and to give rise to distinct differentiated cell types throughout their lifetime [6]. The canonical Wnt signaling program plays a central role in controlling the balance between stemness and differentiation in several adult stem cell niches [15], including the ovary [7]. Accordingly, aberrant Wnt signaling is associated with pathological conditions like cancer [15]. 

Wnt proteins comprise a group of evolutionary conserved, lipid-modified glycoproteins [16] that operate at both short and long distances in order to regulate programs involved in proliferation, differentiation and stemness [15,17]. In absence of canonical Wnt ligands, intracellular β-catenin levels are regulated by the formation of a multiprotein “destruction complex” encompassing protein phosphatase 2A (PP2a), glycogen synthase kinase 3 (GSK3β) and casein kinase 1α (CK1α), and the scaffold proteins adenomatous polyposis coli (APC), and AXIN1/2. The destruction complex binds and phosphorylates β-catenin at specific serine and threonine residues, thereby targeting it for ubiquitination and subsequent degradation by the proteasome (Figure 1a). Instead, in the presence of Wnt ligands, co-activation of the Frizzled and LRP5/6 (low-density lipoprotein receptor-related proteins) receptors prevents the formation of the destruction complex, thereby stabilizing intracellular β-catenin and eventually leading to its translocation from the cytoplasm to the nucleus. Here, β-catenin interacts with members of the T-cell specific transcription factor/lymphoid enhancer binding factor (TCF/LEF) family of transcription factors and modulates the expression of a broad spectrum of Wnt downstream target genes regulating stemness, proliferation, and differentiation [15] (Figure 1b).

An illustrative example of the relevance of a tightly controlled Wnt signal regulation is provided by the intestinal stem cell niche, i.e., the crypt of Lieberkühn. At the bottom of the crypt, where the highly proliferative intestinal stem cells (ISC) reside, Wnt signaling is highly active due to signals from the surrounding stromal compartment [18], as also shown by nuclear β-catenin localization in both ISCs and the intercalating Paneth cells. Moving up along the crypt-villus axis, Wnt becomes progressively less active, following a signaling gradient inversely proportional to the differentiation grade of the epithelial lining [19]. In accordance with the central role played by this Wnt gradient, loss of function mutations at the tumor suppressor gene *APC* or gain of function mutations in the β-catenin (*CTNNB1*) oncogene leading to ligand-independent (i.e., constitutive) Wnt activation represent the main initiating events in the vast majority of sporadic colon cancer cases. Hence, the disruption of the homeostatic equilibrium among stemness, differentiation, and proliferation along the crypt-villus axis brought about by constitutive Wnt activation is sufficient to trigger colon cancer development [20].

The functional relevance of the Wnt pathway in controlling stemness, proliferation, and differentiation in organ-specific adult stem cell niches other than the intestinal tract is reflected by the broad spectrum of cancer where its deregulation contributes to tumor initiation and/or progression. Accordingly, there is ample evidence from the scientific literature supporting an important role for Wnt signaling in both the onset and progression of ovarian cancer [7].

## 3. Wnt Signaling in Ovarian Development and Tissue Homeostasis

Mammalian sex determination is a developmental process consisting of two distinct antagonistic genetic pathways allowing XX or XY undifferentiated gonads to differentiate into two different organs, namely the testis and the ovary [22]. The SRY-SOX9-FGF9 pathway supports testis development, while the RSPO1-Wnt-β-catenin-FOXL2 network promotes ovarian determination [22] (Figure 2).

Before sex determination, the undifferentiated gonad is composed of the coelomic epithelium, together with germ and mesenchymal cells. Here, both the Wnt signaling activators Wnt4 and R-spondin 1 (RPSO1) are important regulators of proliferation of the coelomic epithelium, as indicated by ablation of both *Rspo1* and *Wnt4* leading to reduced numbers of coelomic epithelial cells in XX and XY gonads and, consequently, to hypoplastic testis in XY mutant gonads [23].

During XY sex determination, the transcription factor sex-determining region Y (SRY) together with Splicing factor 1 (SF1) upregulate SRY-Box 9 (*SOX9*) gene expression. Subsequently, *SOX9* upregulation leads to the differentiation of coelomic epithelium into anti-Müllerian hormone producing Sertoli cells, thereby stimulating testis development [24]. Sertoli cells also secrete FGF9 (Fibroblast Growth Factor 9) thus inhibiting the pro-ovarian Wnt signaling pathway [25]. Furthermore, WT1 (Wilms Tumor 1) and ZNFR3 (Zinc Finger 3) also have been shown to downregulate Wnt signaling during male sexual differentiation [26,27]. Accordingly, genetic ablation of *Znrf3* leads to ectopic Wnt signaling in XY gonads and consequentially in the presentation of a female phenotype [27]. 

In females, both granulosa cells and ovarian surface epithelium (OSE) are derived from the coelomic epithelium. During fetal stages, *Rspo1* is expressed in the mesothelial lining of the coelomic cavity and within the fetal ovary [28], whereas *Wnt4* expression is localized to the gonad medulla and mesonephros between the gonad and the Müllerian duct [29]. Wnt4 and RSPO1 are essential for ovarian differentiation and oogenesis as they suppress *Sox9* expression, stimulate granulosa cell differentiation, and promote female sexual development by sustaining Müllerian duct differentiation [28,30,31]. Genetic ablation of *Wnt4, Rspo1,* or *Ctnnb1* in XX gonads lead to premature differentiation of granulosa cells in fetal stages and consequentially to the abrogation of ovary development at perinatal stages [32]. 

Ng et al. (2014) [33] provided additional evidence highlighting the significance of the role played by Wnt during ovarian development and tissue maintenance and regeneration in adulthood. In this study, *Lgr5*, the marker of the above-mentioned and Wnt-driven intestinal stem cells, was shown to be broadly expressed during ovarian organogenesis, whereas it was restricted to the OSE in neonatal life. Using in vivo lineage tracing, *Lgr5^+^* cells were identified as stem or progenitor cells, able to contribute to the development of the OSE cell lineage, the mesovarian ligament, and the fimbriae. In adult ovaries, *Lgr5*^+^ cells were restricted to the proliferative regions of the OSE and the mesovarian-fimbria junctional epithelium. In the OSE, *Lgr5*^+^ cells are thought to preserve homeostasis and to underlie repair of the epithelial damage after ovulation [33]. Indeed, using a Wnt-reporter mouse model, the complete coelomic epithelium overlying the undifferentiated gonad displayed β-catenin/Tcf mediated LacZ expression gradually reduced to smaller populations during postnatal, pre-puberal, and adult life [34]. Of note, the LacZ^+^ OSE cells were enriched in SP (side population) positive cells, a sub-population of stem-like cells identified by their capacity to efflux the dye Hoechst 33342 by ATP-binding cassette super-family G member 2 (ABCG2) transporter pumps [34], a clinically relevant feature acquired by chemotherapy resistant ovarian CSCs.

Apart from its role during embryonic development of the ovary, Wnt signaling was also shown to be an essential regulator of ovarian homeostasis, fertility, and tumorigenesis. Knock-out of *APC2*, a homologue of the *APC* tumor suppressor gene [35], resulted in the activation of ovarian Wnt signaling and in sub-fertility. The latter was due to disturbed follicular growth and the consequent reduced ovulation rate and corpora lutea formation [36]. Notably, aged *APC2*^-/-^ mice developed granulosa cell tumors (GCT) with comparable histological features and molecular signatures to those of the corresponding human GCTs [36]. 

Overall, the central role played by Wnt in regulating the delicate balance between stemness, proliferation, and differentiation to ensure ovarian tissue homeostasis is reflected by its causal association with ovarian cancer onset and/or progression as discussed in the next section.

## 4. Wnt Signaling in Ovarian Cancer

As mentioned above, *CTNNB1* (β-catenin) mutations are found in 16–54% of endometrioid ovarian cancer cases. Likewise, genetic alterations in other members of the Wnt cascade, such as *APC*, *AXIN1*, and *AXIN2*, have also been detected in this specific ovarian cancer histotype [7,8]. In a conditional *APC* knock-out mouse model, it was shown that constitutive activation of Wnt/β-signaling in Müllerian duct-derived organs (i.e., fallopian tubes, uterus, cervix, and the upper two thirds of the vagina) results in the formation of endometrioid tumors in the oviduct, reminiscent of the corresponding histotype in man. Of note, in the same study the ovarian surface epithelium was unaffected, thus suggesting that the oviduct, rather than the OSE, encompasses the cell of origin of (endometrioid) ovarian cancer [37]. 

In addition to endometrioid ovarian carcinomas, mutations in *CTNNB1* are also found in rare cases of mucinous ovarian cancer [38]. Moreover, both *CTNNB1* and *APC* mutations have also been detected in non-epithelial microcystic stromal tumors (MSTs) of the ovary [39,40,41]. Accordingly, an increased incidence of MSTs has also been reported among patients affected by familial adenomatous polyposis (FAP) due to germline mutations in *APC* [40,41]

Yet, it should be clearly stated that endometrioid tumors represent a notable exception as mutations in Wnt-related genes are in general extremely rare in any other ovarian cancer histotype [7]. However, even in the absence of specific mutations, Wnt signaling has been reported to be frequently activated in the more common serous histotype as indicated by nuclear and cytoplasmic β-catenin subcellular localization [9,10,11,12]. In addition, expression profiling data have confirmed the frequent activation of Wnt signaling in ovarian cancer at large [42,43]. In particular, transcriptome analysis of ascites-derived ovarian cancer cells and tumor-associated macrophages (TAMs) has revealed that both canonical and non-canonical Wnt ligands (i.e., *WNT7A*, *WNT2A*, *WNT5A*, *WNT9A*) are expressed in tumor cells, whereas *LRP* and *FZD* are common to both tumor cells and TAMs [43]. 

Pangon et al. (2016) took advantage of the Cancer Genome Atlas (TCGA) to show that the oncogene *JRK* (jerky) is overexpressed in 15% of ovarian cancers in association with increased expression of canonical Wnt target genes [44]. JRK directly interacts with the β-catenin transcriptional complex, thereby stabilizing the β-catenin/TCF complex and ultimately resulting in increased β-catenin transcriptional activity and cell proliferation. Consistent with this, depletion of JRK in cancer cell lines repressed expression of β-catenin target genes and reduced cell proliferation [44]. 

More recently, noncoding RNAs (ncRNAs) have emerged as important post-translational regulators of Wnt-associated gene expression in ovarian cancer (Table 1). By using orthotopic mouse models of ovarian cancer, it was demonstrated that β-catenin plays a key role in the formation of metastasis by controlling the endoribonuclease Dicer, a key component of the microRNA (miR)-processing machinery. β-catenin directly targets Dicer, thereby downregulating multiple miRNAs including the miR-29 family known for its role as a negative EMT regulator. Silencing of β-catenin or overexpression of Dicer or mi-R29 in metastatic ovarian cancer cells reduced their migratory capacity, and attenuated metastasis formation upon β-catenin knockdown in orthotopic mouse models [45]. Of note, reduced expression of miR-29 is associated with ovarian cancer progression and strongly correlated with poor survival [46]. 

Several other miRs have been demonstrated to impact migration, invasion, and cancer progression via Wnt signaling in ovarian cancer [47,48,49,50,51,52,53,54,55,56,57,58,59]. Interestingly, miR-939 has been suggested to function as a tumor promotor by regulating Wnt signaling through direct suppression of the previously discussed *APC2* tumor suppressor [48]. 

Next to miRs, several long non-coding RNAs (lncRNAs) have been described to play a causative role in Wnt-associated cell proliferation, EMT, and chemotherapy resistance in ovarian cancer [60,61,62,63,64,65]. Table 1 summarizes the data relative to gene and non-coding RNA alterations leading to Wnt signaling activation in ovarian cancer.

Apart from the above alterations in genes and non-coding RNAs, Wnt signaling activation in ovarian cancer might result from additional alternative epigenetic mechanisms, either cell-autonomous or induced by the tumor microenvironment. Epigenetic alterations leading to autocrine overexpression of Wnt ligands [72,73], receptors [74], and/or of other Wnt agonists like *FRAT1* [12] or *PYGO2* [75], or to the inhibition of antagonists such as the secreted frizzled receptors proteins (sFRP) and Dickkopf (DKK1) [14,76,77,78] have been reported in the literature. Likewise, paracrine secretion of Wnt-activating cues was observed from either the stroma surrounding the primary ovarian cancer, or from ascites in the case of late-stage disease. Several components of ovarian cancer ascites, known to be associated with shorter progression free survival [79,80], have been previously implicated in promoting Wnt signaling: leptin [81,82], urokinase-type plasminogen activator receptor (uPAR) [83], and macrophage migrating inhibitory factor (MIF) [84]. These soluble factors may act by activating the Wnt pathway in disseminated ovarian cancer cells present in ascites. Two additional ascites factors, namely osteoprotegerin (OPG) [85] and interleukin 8 (IL-8) [86], are in fact downstream Wnt targets and could serve as markers of Wnt signaling activity in ascites. In addition, β-1 integrin-mediated adhesion to the peritoneal mesothelium, a key step in the route to ovarian cancer metastasis, activates β-catenin signaling [87]. Of note, it has recently been shown that extracellular vesicles such as exosomes play a critical role in long-distance transmission of morphogens and in particular in Wnt signaling [88]. In the context of ovarian cancer ascites, exosomes may represent a stable source of paracrine Wnt signals [89]. The role of exosomes will be discussed at more length later on in this review. 

## 5. Wnt Signaling in Ovarian Cancer Stem Cells, EMT, and Therapy Resistance

After diagnosis, tumor debulking surgery followed by carboplatin- and paclitaxel-based chemotherapy represent the standard first line therapy for high grade serous ovarian cancer patients. Although at this stage the primary response to chemotherapy is extremely efficient, most patients relapse and develop metastases locally and at distant organ sites [90]. This is mainly due to sub-populations of tumor cells likely to have acquired stem cell features (CSCs) through EMT and, consequently, the EMT-associated chemo-resistance [5]. In 2005, it was shown for the first time that the aggressiveness of human ovarian cancer results from alterations in stem and progenitor-like cells in the ovary [91]. Moreover, this study demonstrated that the small subpopulation of stem-like, tumor-propagating ovarian cancer cells were earmarked by expression of cluster of differentiation 44 (CD44) and other stem cell and EMT markers such as *KIT* (CD117), *SCF* (stem cell factor), *SLUG* (*SNAIL2*), and *VIM* (vimentin) [91]. After this initial report, several cell surface antigen markers have been identified which allow enrichment of ovarian CSCs from immortalized cell lines, primary tumors, and ascitic fluids: CD133, CD24, CD44, CD177, aldehyde dehydrogenase 1 (ALDH1), and SP [5]. Notably, active Wnt signaling has been shown to play a key role in the regulation and maintenance of ovarian cancer stemness [51,92,93]. 

Ovarian cancer follows a unique pattern of metastasis formation where, unlike many other cancer types, no anatomical barrier exists between the primary site and the abdominal cavity, thus greatly facilitating the dissemination of exfoliated malignant cells. In particular, disseminated ovarian cancer cells secrete vascular permeability factors and can block lymphatic drainage leading to accumulation of ascites fluid within the peritoneal cavity [94]. These malignant ascites provide a favorable tumor microenvironment (TME) enriched in secreted inflammatory cytokines [79], growth factors [95], and extracellular macromolecules (collagen, fibronectin, and laminin) [96]. In this environment, tumor cells form multicellular aggregates enriched in cancer stem/progenitor cells, the so-called ‘spheroids’, which eventually implant on the mesothelial lining of the peritoneum [97] (Figure 3). The attachment of these floating spheroids to the peritoneal lining and associated organs represents the major route for metastasis formation in ovarian cancer [98] where, as observed in other epithelial cancers, EMT was shown to play a key role [99]. Interestingly, although hematogenous spread is generally thought to play a relatively minor role in metastasis formation in ovarian cancer, it has recently been demonstrated in a parabiosis mouse model [100]. In this study, two mice, one of which intraperitoneal transplanted with ovarian cancer cells, were surgically connected to share blood supply. The development of ovarian cancer in the cancer-free animal likely results from hematogenous spread [100]. Likewise, circulating tumor cells have been identified in peripheral blood from ovarian cancer patients [101]. 

Overall, the naturally occurring spheroids in ascites are likely to underlie metastatic disease in ovarian cancer patients. In the next sections, we will discuss the current experimental evidence on the role of Wnt signaling in eliciting EMT and chemo-resistance in high grade serous ovarian cancer.

### 5.1. Wnt Signaling and EMT in Ovarian Cancer

EMT is a reversible developmental program exploited by cancer cells to reversibly switch from an epithelial phenotype with apical-basal polarity and cell–cell adhesions, to a more motile mesenchymal state with spindle like morphology and front-back-end polarity [102]. Next to the motility and invasive features characteristic of the mesenchymal state, EMT is functionally associated with the acquirement of stem-like features, resistance to therapy, and immune suppression [103,104,105]. Last, the capacity of cells undergoing EMT to revert to an epithelial state by mesenchymal-to-epithelial transition (MET) is rate-limiting to allow the stem- and mesenchymal-like migrating CSCs to regain proliferative and epithelial features essential to colonize the metastatic site [102,106]. Various signaling pathways are involved in EMT, including transforming growth factor β (TGF-β), Notch, and Wnt/β-catenin. Activation of the Wnt/β-catenin pathway has been shown to be an important regulator of EMT in many different types of cancers [106,107,108], including ovarian cancer [109,110,111,112]. In this context, ovarian cancer cell lines with a high SNAIL to E-cadherin ratio, are characterized by enhanced CSC-like, motile, and therapy-resistant features when compared with epithelial ovarian cancer cell lines. Accordingly, *SNAIL* knockdown reversed the malignant properties and tumor burden of the more mesenchymal ovarian cancer cell lines in xenograft models [111]. *SNAIL* and other EMT transcription factors (EMT-TFs) have been shown to activate expression of the *GOLPH3* (Golgi phosphoprotein 3) gene, encoding for an oncoprotein frequently upregulated in ovarian cancer tissues and cell lines, through Wnt/β-catenin signaling activation [112]. Induction of EMT and the consequent acquisition of migratory and invasive cellular features downstream of Wnt activation have also been demonstrated in ovarian carcinomas where *IQGAP2*, a Wnt antagonist, is frequently silenced by DNA methylation [110]. Last, cyclin G2, an unconventional cyclin that opposes cell cycle progression and inhibits EMT, acts as a tumor suppressor in ovarian cancer by inhibiting Wnt/β-catenin signaling [109].

More recently, it has been suggested that, rather than being a binary process with fully opposing epithelial and mesenchymal phenotypes, EMT generates hybrid E/M cancer cells displaying both epithelial and mesenchymal characteristics [113,114]. Indeed, similar to the normal ovarian surface epithelial (OSE) cells previously shown to display both epithelial and mesenchymal characteristics and a remarkable phenotypic plasticity during post-ovulatory repair, double positive E-cadherin and vimentin cells have been observed in ovarian cancers [115]. Accordingly, different intermediate EMT states have been identified in ovarian cancer cell lines [116] and ascites-derived spheroids [117]. Here, ovarian cancer cells in hybrid E/M states were shown to exhibit stem-like features, anoikis resistance, and increased migration and invasion when compared with the fully epithelial and mesenchymal states [116,117,118]. 

Overall, it is yet unclear whether the hybrid E/M cells represent a ‘metastable’ cell population or are cells captured in a time frame during the transition between the epithelial to mesenchymal states. The elucidation of the complex network of intrinsic and extrinsic mechanisms underlying EMT during metastasis formation and the role of Wnt signaling therein represents an important future research challenge. In the next section, the current knowledge on the role of Wnt signaling in resistance to chemotherapy in ovarian cancer will be discussed. 

### 5.2. Wnt Signaling and Therapy Resistance in Ovarian Cancer

As mentioned above, chemotherapy is extremely efficient in the first-line treatment of primary ovarian cancers although it inevitably leaves behind chemo-resistant CSCs likely to underlie relapse and metastasis in distant organ sites [90]. Wnt signaling has been associated with resistance to chemotherapy in different tumor types including ovarian cancer [119]. 

Chemo-resistance can be acquired through a broad spectrum of molecular and cellular mechanisms such as the upregulation of ATP-binding cassette (ABC) transporter pumps, the activation of EMT, and the exosome-mediated transport of molecules controlling a broad spectrum of pathways underlying drug resistance [120]. ABC transporters have indeed been shown to be expressed in ovarian cancer usually in association with cancer stemness and poor prognosis [121,122]. Notably, upregulation of the ABCG2 transporter pump and Wnt signaling activation downstream of cKIT mediate the onset of resistance to cisplatin and paclitaxel in ovarian CSCs [13]. In the same study, *ABCG2* expression and chemo-resistance to both cisplatin and paclitaxel could be reversed by β-catenin siRNA knockdown, once again highlighting the central role of Wnt signaling in these processes [13]. 

Another well-established mechanism underlying therapy resistance in ovarian cancer, as also mentioned in the previous section, is represented by EMT [99]. Su et al. (2010) showed that *SFRP5* (secreted frizzled-related protein 5), a well-known Wnt and EMT antagonist, is frequently downregulated in ovarian cancer by epigenetic silencing through promoter hypermethylation [123]. Accordingly, restoration of *SFRP5* expression inhibits Wnt signaling and EMT thus sensitizing ovarian cancer cells to chemotherapy. Activation of the EMT-TF *TWIST* and of *AKT2* signaling play key roles downstream of *SFRP5* silencing [123]. 

In addition to the above-mentioned cell-autonomous mechanisms, ascites also forms a unique tumor microenvironment likely to contribute to therapy resistance [124]. Malignant ascites provides a favorable tumor microenvironment consisting of cellular and non-cellular components, each likely to play a role in the development of resistance to carboplatin- and paclitaxel-based therapy. Among these, cancer associated fibroblasts (CAFs) represent an important component of ovarian cancer ascites [124]. CAFs are a subpopulation of fibroblasts capable of affecting tumor progression, dissemination, and therapy response through signaling to tumor cells and/or remodeling of the extracellular matrix (ECM) [125]. Recently, Ferrari et al. (2019) demonstrated that Dickkopf-3 (DKK3), the stromal expression of which is strongly associated with aggressive ovarian cancer, promotes CAFs’ aggressive behavior by enhancing Yes-associtated protein/transcriptional co-activator with PDZ-binding motif (YAP/TAZ) activity through Wnt/β-catenin signaling [126]. From a mechanistic perspective, DKK3 destabilizes the Wnt-antagonist Kremen, leading to increased LRP6 localization at the cell membrane. This in turn stabilizes YAP/TAZ and β-catenin levels leading to more global gene expression changes enhancing cancer stemness, malignant progression, and metastasis [126]. Other ascites cellular components such as macrophages have also been shown to take part in tumor progression and the development of therapy resistance. Ragahvan et al. (2019) showed that Wnt signaling participates in a bidirectional ovarian CSC-macrophage interaction [92]. By taking advantage of hetero-spheroids composed of macrophages and ovarian cancer cells in close contact with each other, it was shown that Wnt signaling, activated by secretion of the Wnt5b ligand from macrophages, led to an increase of the ovarian CSC compartment (ALDH^+^) and to the enhancement of the immune-suppressive characteristics of the macrophages. Likewise, Wnt5b knockdown in macrophages resulted in a loss of the ALDH^+^ ovarian CSC fraction. Most importantly, the hetero-spheroids were less sensitive to chemotherapeutics and were more invasive in in vitro assays [92]. Hence, macrophage-initiated Wnt activation is likely to play a central role in ovarian cancer stemness maintenance and in therapy resistance. 

Notwithstanding more recent advances in chemotherapy (e.g., intraperitoneal delivery of cytotoxic drugs and the introduction of novel, more targeted agents such as bevacizumab and imatinib) [127,128,129], less than 30% of advanced ovarian cancer patients survive longer than five years after diagnosis [1]. Therefore, there is urgent need for novel therapeutic strategies based on improved understanding of the molecular and cellular mechanisms underlying dissemination and metastasis formation by ovarian cancer cells in the peritoneal cavity and their acquisition of dormant and chemo-resistant properties. Recently, the role played by extracellular vesicles and in particular by exosomes in tumor progression, dissemination, and resistance to therapy has opened new avenues in basic and translational cancer research. In the next section we will present and discuss the current knowledge on exosomes in ovarian cancer, especially in the context of intra-abdominal ascites and of long-range Wnt signaling activation.

## 6. Exosomes and Wnt Signaling in Ovarian Cancer Ascites

Malignant ascites provides a favorable tumor microenvironment and consists of a heterogeneous mixture of cells and secreted factors that modulate cancer cell behavior during tumor progression, metastasis formation, and acquirement of chemo-resistance. As mentioned, Wnt ligands are modified lipids and are therefore highly hydrophobic, thereby limiting their ability for extracellular diffusion [16]. Recently however, studies have shown that Wnts can be transported across tissues by exosomes [88,130]. In the following paragraphs we will highlight the current knowledge on the role played by exosomes in ovarian cancer ascites as a putative mechanism to activate Wnt signaling over long-range distances both in establishing pre-metastatic niches in the peritoneal cavity and in preserving stemness in disseminated cancer cells.

### 6.1. Exosomes

Exosomes are small extracellular vesicles ranging in diameter from 30 to 100 nm that are secreted by most eukaryotic cells. Secreted exosomes are important mediators in cell–cell communication as they carry molecules such as microRNAs, mRNAs, and both membrane-bound and secreted proteins [131]. Exosomes are thought to facilitate tumor survival and progression by stimulating angiogenesis and tumor growth, suppressing immune responses, remodeling of the extracellular matrix, promoting metastasis formation either directly and/or through the establishment of premetastatic niches [131] (Figure 3). Numerous studies have demonstrated the presence of exosomes in ovarian cancer cell line cultures, and in patient-derived serum and ascites [132,133,134,135]. Notably, it has been shown that active Wg (Wingless) and Wnt3a ligands are membrane-bound in exosomes from Drosophila and human cells, respectively [88]. Moreover, macrophage-derived and exosome-packaged Wnts are rate-limiting for the regenerative response of intestine intestinal stem cells after radiation [130]. In relation to cancer, fibroblast-derived exosomes carrying Wnt ligands increase cell migration and metastasis formation in breast cancer [136]. Hu et al. (2019) recently found that exosomes derived from stromal fibroblasts contain Wnt ligands capable of eliciting the de-differentiation of colon cancer cells into therapy resistant CSCs [137]. Alternatively, activation of Wnt signaling in target cells has been shown to occur by exosomes encompassing β-catenin in their cargo. Here, both 14-3-3 proteins and β-catenin were encompassed in the extracellular vesicles. 14-3-3 proteins bind to dishevelled segment polarity protein 2 (Dvl-2) and GSK3β thereby interfering with β-catenin phosphorylation and stimulating Wnt signaling [138]. 

Although to date no evidence has been presented supporting the presence of exosomes encompassing active Wnt ligands in ovarian cancer ascites, differential expression analysis of ovarian cancer exosomes compared with those from normal OSE cells indicate a potential involvement of miRNAs known to target the Wnt signal transduction pathway [139]. Moreover, recently it has been demonstrated that exosomes isolated from a highly invasive ovarian cancer cell line promote metastasis in vivo compared to exosomes from cells with low invasive capacity [140]. Quantitative proteomic analysis of tumor tissues of the mice treated with exosomes derived from these two different cell lines revealed a potential role for Wnt signaling in the role played by exosomes in tumor growth and metastasis in vivo [140]. Also, as discussed here below, ovarian cancer exosomes containing the Wnt target and transmembrane protein CD44 have been shown to participate in the formation of pre-metastatic niches [141]. 

### 6.2. Pre-Metastatic Niche

Ovarian carcinomas spread through the shedding of clusters of tumor cells from the primary lesion into the peritoneal cavity. In this context, the key event in metastatic seeding is the mesothelial adhesion of ovarian cancer cells in the intraperitoneal cavity. The establishment of premetastatic niches is thought to be required for disseminating cancer cells to engraft at the distant site [142]. Premetastatic niches comprise of a specialized and favorable micro-environment that facilitates colonization and promotes survival and outgrowth of disseminated tumor cells [142] (Figure 3). The relevance of the formation of pre-metastatic niches in ovarian cancer has been proposed by several studies [141,143]. Lee et al. (2019) demonstrated that inflammatory factors secreted by ovarian cancer cells mobilize neutrophils and stimulate them to create chromatin webs called ‘neutrophil extracellular traps’ (NETs) in the omentum in both tumor-bearing mice (before metastasis occurs) and in early-stage ovarian cancer patients. The NETs can sequentially capture ovarian cancer cells and thereby promote metastasis formation. Reversely, inhibiting NET formation abrogated omental colonization [143].

Next to NETs, ovarian cancer exosomes have also been shown to participate in the establishment of a pre-metastatic niche by alternative mechanisms. First, *MMP1* mRNA has been found in extracellular vesicles derived from ovarian cancer cell lines and ascites from ovarian cancer patients that promotes apoptotic cell death of the mesothelial cells, thus resulting in the destruction of the peritoneal barrier [144]. In addition, ovarian cancer cells’ exosomes encompassing the cell-surface glycoprotein CD44 can transfer it to peritoneal mesothelial cells and induce their reprogramming by EMT activation. The modified mesothelium facilitates ovarian cancer invasion and metastasis formation [141]. Of note, CD44 is a major Wnt target gene in the intestinal epithelium [145] and is essential for Wnt induction during colon cancer progression [146], thus suggesting yet another functional link between Wnt signaling and ovarian cancer exosomes in pre-metastatic niche formation. 

To interfere with the interaction between disseminated ovarian cancer cells and the exosome-receiving mesothelial cells, De la Fuente et al. (2015) developed a metastatic trap (M-Trap) [147]. By embedding exosomes purified from ovarian cancer patient ascites on a 3D scaffold, the authors showed that the M-Trap device was able to capture ovarian cancer cells in a mouse model of ovarian cancer. This led to a more focalized disease and an increase in survival rate [147]. These results lay the foundation for future clinical approaches to improve treatment of ovarian cancer patients with malignant ascites [147]. 

Overall, notwithstanding that treatment of advanced stage ovarian cancer patients still represents a major clinical challenge, recent advances in our understanding of the mechanisms underlying ovarian cancer ascites formation and the role they play in metastasis formation in the peritoneal cavity are of good auspices for the future. Exosomes in particular, may represent powerful tools in early diagnosis and treatment [135]. As for the latter, targeted exosome ablation or inhibition of exosome secretion may affect tumor progression or therapy resistance (Figure 4). In this scenario, the Wnt signaling pathway may also represent a relevant therapeutic target. In the next paragraph, treatment options based on targeting of Wnt signaling in ovarian cancer will be discussed.

## 7. Targeting Wnt in Ovarian Cancer: Opportunities for Treatment?

During the last decade, the therapeutic response rate of ovarian cancer patients has improved through optimization of chemotherapy strategies, their intraperitoneal administration, and the introduction of targeted therapies [127,128,129]. However, despite these developments, the overall survival of ovarian cancer patients has not significantly improved [1]. Because of the role played in cancer stemness and in therapy resistance, the Wnt signaling pathway forms a candidate target for therapeutic intervention as different segment of this cascade are suitable for therapeutic targeting (Figure 4; Table 2). 

Although R-spondins (RSPO) are unable to initiate Wnt signaling, they can, by binding to leucine-rich repeat-containing G-protein coupled receptors (LGR) enhance responses to low-dose Wnt proteins [148]. Functional RSPOs have been found in multiple human tumor types and anti-RSPO monoclonal antibodies shown to reduce the tumorigenicity of cancer cells in patient-derived tumor xenograft models of several malignancies including ovarian cancer [149]. Porcupine (PORCN) inhibitors form another relevant target to inhibit Wnt signaling. The acetyltransferase PORCN is responsible for post-translational modifications of Wnt proteins essential for the transport, secretion, and activity of the ligands. WNT974 is a selective PORCN inhibitor that has been shown to exert cytostatic effects on ascites-derived ovarian cancer cells as a consequence of Wnt signaling inhibition [150]. When combined with the conventional chemotherapeutic drug carboplatin, WNT974 administration led to increased cytotoxic effects and cell cycle arrest in ascites samples when compared with single drug treatments [150]. 

The FDA (Food and Drug Administration)-approved anti-helminth compound niclosamide represents yet another powerful Wnt inhibitor shown to repress ovarian CSC growth through downregulation of both the disheveled protein DVL2 and the surface receptor LRP6 [151]. Next to Wnt, niclosamide targets additional signaling pathways known to play a role in cancer stemness, including Notch, mTORC1, and Stat3 [152].

Besides the above mentioned Wnt targets and inhibitory compounds, inhibition of Wnt ligands secretion, inactivation of the extracellular portion of Frizzled receptors, and interference with the TCF/β-catenin complex represent additional and presently under investigation strategies [153] (Figure 4; Table 2). 

Currently, different Wnt inhibitors are being evaluated in clinical trials for different cancer types including ovarian cancer. As a notable example, Ipafricept is a recombinant fusion protein that competes with the FZD8 receptor for its ligand thereby antagonizing Wnt signaling. Ipafricept reduces cancer stem cells, promotes differentiation, and synergizes with taxanes in ovarian cancer xenografts. More recently, a phase 1B trial was conducted with ipafricept in combination with carboplatin and paclitaxel in patients with recurrent platinum-sensitive ovarian cancer [154]. Unfortunately, although generally well-tolerated by patients, bone toxicity at efficacy doses limited ipafricept treatment [154]. Nonetheless, other Wnt inhibitors targeting PORCN and β-catenin are now being tested in clinical trials in different tumor types [155]. 

## 8. Conclusive Remarks

In conclusion, a considerable body of evidence supports the relevance of the role played by Wnt signaling in ovarian cancer stemness, progression to malignancy, and resistance to chemotherapy. Notwithstanding the potential and innovative therapeutic strategies currently in development to specifically target the Wnt pathway, plasticity of cancer cells still represents an escape mechanism leading to therapy resistance. Moreover, because of Wnt’s essential role in tissue homeostasis and regeneration upon damage, its inhibition is likely to result in adverse events. Therefore, the identification and elucidation of the complex network of intrinsic and extrinsic mechanisms driving ovarian cancer progression and therapy resistance represent the major future research challenge in the translation of the fundamental understanding of metastasis and therapy.

## Figures and Tables

**Figure 1 jcm-08-01658-f001:**
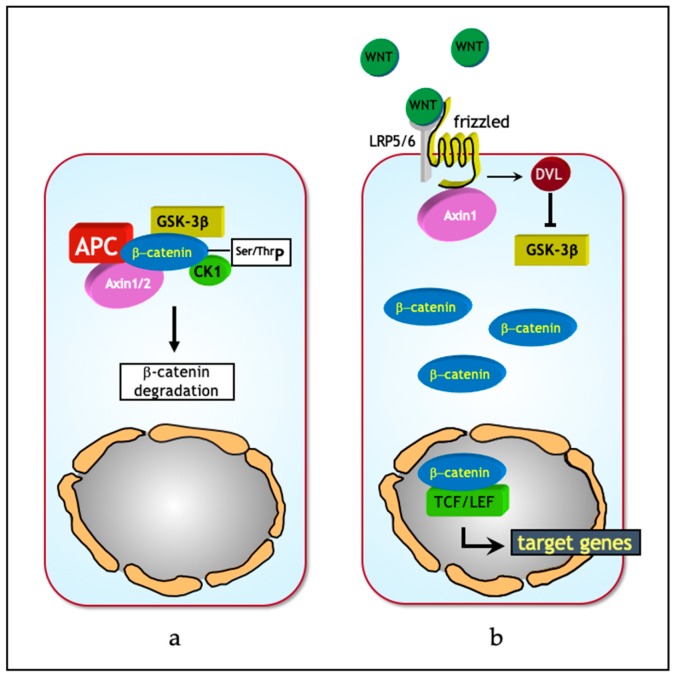
The Wnt/β-catenin signal transduction pathway in homeostasis. (**a**) In the absence of Wnt ligands, intracellular β-catenin levels are controlled by a destruction complex encompassing protein phosphatase 2A (PP2a), glycogen synthase kinase 3 (GSK3β) and casein kinase 1α (CK1α), adenomatous polyposis coli (APC), and AXIN1/2. This complex binds and phosphorylates β-catenin at serine and threonine residues, thereby targeting it for ubiquitination and proteolytic degradation by the proteasome. (**b**) In presence of Wnt, co-activation of the Frizzled and low-density lipoprotein receptor-related protein 5/6 (LRP5/6) (low-density lipoprotein receptor-related proteins) receptors prevents the formation of the destruction complex leading to the stabilization and consequent translocation of β-catenin from the cytoplasm to the nucleus. Here, β-catenin interacts with members of the T-cell specific transcription factor/lymphoid enhancer binding factor (TCF/LEF) family of transcription factors and modulates the expression of a broad spectrum of Wnt downstream target genes. DVL – disheveled. Adapted from [21].

**Figure 2 jcm-08-01658-f002:**
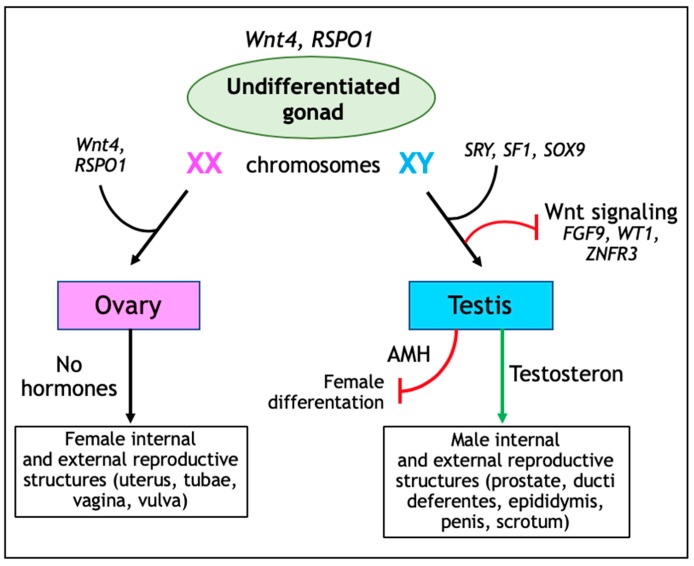
Schematic view of sex determination. In the undifferentiated gonad both Wnt4 and RSPO1 (R-spondin 1) are important regulators in particular for the proliferation of the coelomic epithelium. In XX gonads, expression of Wnt4, and RSPO1 leads to ovarian differentiation and oogenesis as they suppress *Sox9* expression, stimulate granulosa cell differentiation, and promote female sexual development by sustaining Müllerian duct differentiation. In XY gonads male reproductive organs are determined by the expression of sex-determining region Y (SRY) together with Splicing factor 1 (SF1) leading to upregulation of *Sox9* gene expression. In addition, Fibroblast Growth Factor 9 (FGF9), Wilms Tumor 1 (WT1), and Zinc Finger 3 (ZNFR3) inhibit the pro-ovarian Wnt signaling pathway. Also, anti-Müllerian hormone (AMH) prevents the development of the Müllerian duct into female reproductive organs.

**Figure 3 jcm-08-01658-f003:**
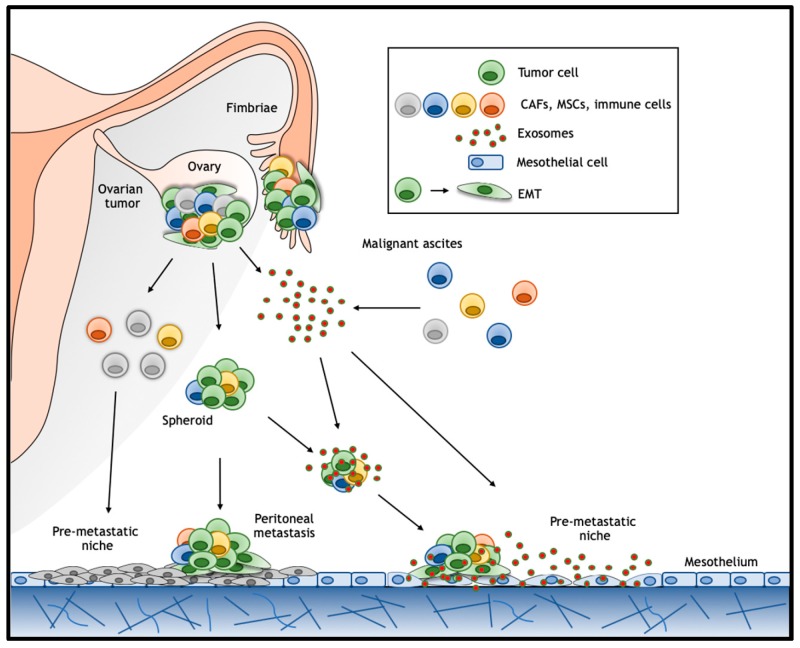
Model for peritoneal metastasis formation in ovarian cancer. Ovarian cancer follows a unique pattern of metastasis formation, where no anatomical barrier exists between the primary site and the abdominal cavity. Multicellular aggregates enriched in cancer stem/progenitor cells, the so-called spheroids, detach from the primary tumor and eventually implant on the mesothelial lining of the peritoneum. EMT was shown to play a key role facilitating the acquisition of stem-like features, anoikis resistance, and increased migration and invasion. The establishment of premetastatic niches composed of several cell populations, including tumor-associated neutrophils, is thought to be required for disseminating carcinoma cells to engraft at the distant site. Exosomes in ovarian cancer ascites have been proposed as a putative mechanism to facilitate long-range distance cell–cell communication thereby establishing both pre-metastatic niches in the peritoneal cavity and preserving stemness in disseminated cancer cells. CAFs: cancer associated fibroblasts; MSCs: mesenchymal stem cells.

**Figure 4 jcm-08-01658-f004:**
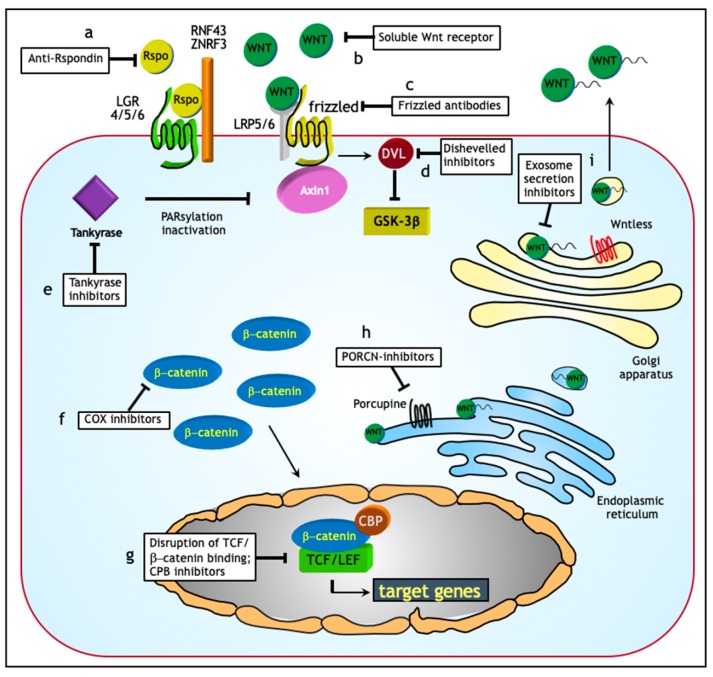
Therapeutic targets for the inhibition of Wnt signaling. (**a–c**) Wnt soluble receptors, anti-R-spondin antibodies, and antibodies directed against Frizzled receptors impair the ligand/receptor interaction and prevent downstream signaling. (**d**) Disheveled inhibitors block Wnt signaling by interfering with the Frizzled/Disheveled interaction. (**e**) Tankyrase activates Axin through PARsylation. Tankyrase inhibition increases Axin levels thus stimulating the formation of the β-catenin destruction complex and reducing the intracellular β-catenin pool. (**f**) cyclooxygenase (COX) inhibitors increase ubiquitination and proteasomal degradation of β-catenin. Next, COX2 inhibition leads to reduced levels of prostaglandin E2 (PGE2) known to positively affect Wnt signaling. (**g**) Disruption of its interaction with TCF inhibits β-catenin-mediated transcriptional activity. CREB-binding protein (CBP) inhibitors instead interfere with the interaction between TCF/LEF and CBP thereby reducing transcriptional activity. (**h**) *PORCN*-inhibitors hamper the palmitoylation of Wnt before its extracellular release. (**i**) Exosome secretion inhibitors reduce the transport of biomolecules like active Wnt ligands, RNAs and proteases that contribute to angiogenesis, tumor growth, immune response suppression, the remodeling and degradation of the extracellular matrix (ECM). Additional abbreviatons: RNF43 = RING finger 43; LGR4/5/6 = Leucine-rich repeat-containing G-protein coupled receptor 4/5/6; RSPO = R-spondin; ZRNF3 = Zinc RING finger 3; GSK3β = glycogen synthase kinase 3β; LRP5/6 = LDL Receptor Related Protein 5/6; TCF/LEF = T-cell specific transcription factor/lymphoid enhancer binding factor.

**Table 1 jcm-08-01658-t001:** Gene and non-coding RNA alterations leading to Wnt signaling activation in ovarian cancers.

Gene/ncRNA	Ovarian Cancer Histotype^*^	Mechanism/Target	Reference
*CTNNB1*	Endometrioid.	Oncogenic activation.	[8,38,66,67,68,69,70]
*CTNNB1*	Mucinous.	Oncogenic activation.	[38]
*CTNNB1*	Microcystic Stromal Tumors (MST).	Oncogenic activation.	[39]
*APC*	Endometrioid.	Loss of tumor suppressor function.	[8]
*APC*	Microcystic Stromal Tumors (MST).	Loss of tumor suppressor function.	[40,41]
*AXIN1*	Endometrioid.	Loss of tumor suppressor function.	[8]
*AXIN2*	Endometrioid.	Loss of tumor suppressor function.	[8]
microRNA (miR)-10a)	Granulosa cell tumor.	miR-10a targets *PTEN* and indirectly activates Wnt (and AKT) signaling. Oncogenic activation.	[54]
miR-15b	Epithelial ovarian cancer *.	miR-15b targets *WNT7A* 3’-untranslated region (3’-UTR) and thus inhibits Wnt signaling. Loss of tumor suppressor function.	[50]
miR-16	Epithelial ovarian cancer *.	miR-16 target(s) yet unknown; it inhibits Wnt signaling. Loss of tumor suppressor function.	[56]
miR-21	Epithelial ovarian cancer *.	miR-21 target(s) yet unknown; it activates Wnt signaling. Oncogenic activation.	[57]
miR-27a	Epithelial ovarian cancer *.	mir-27 targets the Wnt antagonist *FOXO1*. Oncogenic activation.	[55]
miR-29	Serous, mucinous, and clear cell ovarian cancer.	miR-29 target(s) yet unknown; it activates Wnt signaling. Oncogenic activation.	[45,46]
miR-92a-1	Epithelial ovarian cancer *.	miR-92a-1 targets the Wnt antagonist Dickkopf 1 *(DKK1)*. Oncogenic activation.	[51]
miR-200c	Epithelial ovarian cancer *.	miR-200c target(s) yet unknown; it inhibits Wnt signaling. Loss of tumor suppressor function.	[47]
miR-214	Epithelial ovarian cancer *.	miR-214 target(s) yet unknown; it inhibits Wnt signaling. Loss of tumor suppressor function.	[53]
miR-219-5p	Epithelial ovarian cancer *.	miR-219-5p targets the EMT transcription factor *TWIST* and inhibits Wnt signaling. Loss of tumor suppressor function.	[52]
miR-654-5p	Epithelial ovarian cancer *.	miR-654-5p targets *CDCP1* and *PLAGL2*. Loss of tumor suppressor function.	[58]
miR-939	Epithelial ovarian cancer *.	miR-939 targets *APC2*. Loss of tumor suppressor function.	[48]
miR-1180	Epithelial ovarian cancer *.	miR-1180 targets *SFRP1*. Loss of tumor suppressor function.	[59]
miR-1207	Epithelial ovarian cancer *	miR-1207 targets *SFRP1*, *AXIN2*, and *ICAT*. Loss of tumor suppressor function.	[49]
HOTAIR ^1^	Epithelial ovarian cancer *.	HOTAIR target(s) unknown; Wnt agonist. Oncogenic activation.	[60]
SNHG20 ^2^	Epithelial ovarian cancer *.	SNHG20 target(s) unknown; Wnt agonist. Oncogenic activation.	[61]
HOXD-AS1 ^3^	Epithelial ovarian cancer *.	HOXD-AS1 targets the Wnt antagonist miR-133a-3p. Oncogenic activation.	[62]
CCAT2 ^4^	Epithelial ovarian cancer *.	Targets unknown; EMT and Wnt agonist. Oncogenic activation.	[63]
MALAT1 ^5^	Epithelial ovarian cancer *.	Targets unknown; Wnt agonist. Oncogenic activation.	[64]
AWPPH ^6^	Epithelial ovarian cancer *.	Targets unknown; Wnt agonist. Oncogenic activation.	[65]
HOXB-AS3 ^7^	Serous ovarian cancer samples; other histotypes.	Targets unknown; Wnt agonist. Oncogenic activation.	[71]

*, histotype not characterized; ^1^, HOTAIR—HOX antisense intergenic RNA; ^2^, SNHG20—small nucleolar RNA host gene 20; ^3^, HOXD-AS1—HOXD cluster antisense RNA 1; ^4^, CCAT2—colon cancer-associated transcript 2; ^5^, MALAT1—metastasis associated lung adenocarcinoma 1; ^6^, AWPPH—associated with poor prognosis of hepatocellular carcinoma; ^7^, HOXB-AS3—HOXD cluster antisense RNA 3.

**Table 2 jcm-08-01658-t002:** Wnt inhibitors in ovarian cancer.

Molecular Targets	Inhibitors	Activity	Reference
Extracellular targeting	Anti-Rspondin	anti-RSPO monoclonal antibodies reduce tumorigenicity of cancer cells in patient-derived ovarian tumor xenograft models.	[149]
	Ipafricept (OMP54F28)	Recombinant fusion protein that competes with the membrane-bound Frizzled 8 (FZD8) receptor for its ligand; leads to tumor regression in combination with taxane in ovarian xenograft models; currently under clinical trial.	[154]
LRP6 inactivation	Salinomycin	Small molecule blocking Wnt induced LRP6 phosphorylation and induces its degradation; leads to repression of EMT in epithelial ovarian cancer.	[156,157]
Dishevelled	3289–8625	Small molecule disrupting the frizzled-disheveled interaction by targeting the PDZ domain; chemo-sensitizes paclitaxel-resistant ovarian cancer cells.	[158]
PORCN	WNT974	Small molecule inhibitors of Wnt acetyltransferase porcupine; increases cytostatic effects on ascites-derived ovarian cancer cells.	[150]
CK1α activation	Pyrvinium	Small molecule that selectively potentiates CK1α kinase activity leading to increased β-catenin phosphorylation; enhances sensitivity to chemotherapy of ovarian cancer cells.	[159,160]
Non-specific or overlapping targets	Niclosamide	Small molecule inhibitor promoting FZD1 endocytosis and suppressing LRP6 expression; inhibits growth and increases cell death in ovarian cancer.	[161,162,163]
	COX-inhibitors	Aspirin lowers the risk of ovarian cancer development; in case of ovarian cancer underlying mechanism yet unknown.	[164]

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
