# Peer review of "Wnt Signaling in Ovarian Cancer Stemness, EMT, and Therapy Resistance"

_jcm, 2019, doi:10.3390/jcm8101658_

Round 1

Reviewer 1 Report

The authors provide a thorough and comprehensive overview of the role of Wnt signaling in ovarian cancer, with a particular focus on cancer stemness, EMT and resistance to therapies. The specific focus of the review is preceded by an exhaustive and helpful introduction on the biochemical and cell biological features of the Wnt pathway, and is followed by a very pertinent assessment of the pathway itself as a therapeutic target in ovarian cancer.

The article is well written and properly structured, and will represent a useful tool for the scientific community interested in Wnt signaling and/or in ovarian cancer, particularly in cancer stem cells. I have no major concerns or criticisms.

Minor Points:

1. I would suggest to modify the title according to the topics emphasized in the review and the specific focus of the Special Issue. Forexample, "Wntsignaling in Ovarian Cancer Stemness, EMT and therapy resistance" would likely be more appropriate.

2. Line 251. "... most patients relapse and develop metastases at distant organ sites". As the authors illustrate very well in the following paragraph, high-grade serous ovarian cancer commonly disseminate locally into the peritoneal cavity. Therefore, the sentence should be modified accordingly.

Author Response

The authors provide a thorough and comprehensive overview of the role of Wnt signaling in ovarian cancer, with a particular focus on cancer stemness, EMT and resistance to therapies. The specific focus of the review is preceded by an exhaustive and helpful introduction on the biochemical and cell biological features of the Wnt pathway, and is followed by a very pertinent assessment of the pathway itself as a therapeutic target in ovarian cancer.

The article is well written and properly structured, and will represent a useful tool for the scientific community interested in Wnt signaling and/or in ovarian cancer, particularly in cancer stem cells. I have no major concerns or criticisms.

We thank reviewer #1 for his/her kind words. 

Minor Points:

I would suggest to modify the title according to the topics emphasized in the review and the specific focus of the Special Issue. For example, "Wnt signaling in Ovarian Cancer Stemness, EMT and therapy resistance" would likely be more appropriate.

We have modified the title to highlight the role of Wnt signaling in stemness, EMT and therapy resistance in ovarian cancer.

2. Line 251. "... most patients relapse and develop metastases at distant organ sites". As the authors illustrate very well in the following paragraph, high-grade serous ovarian cancer commonly disseminate locally into the peritoneal cavity. Therefore, the sentence should be modified accordingly.

We slightly modified the sentence changing 'at distant organ sites' to locally and at distant organ sites'. 

Reviewer 2 Report

Overall this is a well written review on a complex topic. It may benefit from a schematic outline at the begging of the review of how the Wnt signalling is involved in ovarian cancer although I appreciate this may be difficult to represent. The English language does not require editing except for minor typos.

Author Response

Overall this is a well written review on a complex topic. It may benefit from a schematic outline at the begging of the review of how the Wnt signalling is involved in ovarian cancer although I appreciate this may be difficult to represent. The English language does not require editing except for minor typos.

We thank for reviewing this manuscript and his/her kind words.